# Macular Alterations in a Cohort of Caucasian Patients Affected by Retinitis Pigmentosa

**DOI:** 10.3390/diagnostics14212409

**Published:** 2024-10-29

**Authors:** Marcella Nebbioso, Elvia Mastrogiuseppe, Eleonora Gnolfo, Marco Artico, Antonietta Moramarco, Fabiana Mallone, Samanta Taurone, Annarita Vestri, Alessandro Lambiase

**Affiliations:** 1Department of Sense Organs, Faculty of Medicine and Odontology, Sapienza University of Rome, p.le A. Moro 5, 00185 Rome, Italy; elvia.mastrogiuseppe@uniroma1.it (E.M.); eleonora.gnolfo@uniroma1.it (E.G.); marco.artico@uniroma1.it (M.A.); antonietta.moramarco@uniroma1.it (A.M.); fabiana.mallone@uniroma1.it (F.M.); alessandro.lambiase@uniroma1.it (A.L.); 2Department of Moviment, Human and Health Sciences, University of Rome, Foro Italico, 00135 Rome, Italy; t.samanta@yahoo.it; 3Department of Public Health and Infectious Disease, Sapienza University of Rome, p.le A. Moro 5, 00185 Rome, Italy; annarita.vestri@uniroma1.it

**Keywords:** cystoid macular edema, ellipsoid zone, epiretinal membranes, hereditary retinal dystrophies, lamellar macular hole, retinitis pigmentosa, spectral-domain optical coherence tomography (SD-OCT), subfoveal choroid thickness, vitreomacular traction

## Abstract

Objectives: Our objective was to investigate the prevalence of macular complications detected by spectral-domain optical coherence tomography (SD-OCT) in a large Caucasian cohort of RP patients, highlight the major alterations in chorioretinal structure, and compare the macular structural changes in eyes affected by retinal dystrophies with those in healthy controls. Methods: This was an observational, retrospective, and cross-sectional study. Three hundred and seven patients with RP were consecutively enrolled and underwent clinical assessment. In particular, SD-OCT images were used to ascertain the morphology of the posterior pole of patients with RP by evaluating the residual ellipsoid zone (EZ), the volume and thickness of the outer nuclear layer (ONLT), and subfoveal choroid thickness (SCT). At the same time, the pathological finding that the patients’ vision was reduced under treatment was analyzed. Results: A total of 436 eyes of 218 patients with RP were studied. Considering all of the eyes studied, 103 had cystoid macular edema (CME) (23.62%), 123 (28.21%) had vitreomacular traction (VMT), and 199 (45.75%) had epiretinal membranes (ERMs). There were also 12 (2.75%) cases of lamellar macular holes (LMHs), of which 3 (1.38% of all patients) cases were bilateral. Only 137 eyes (31.42%) did not have the above-mentioned alterations. SCT was significantly reduced compared to that of the control group (193.03 µm ± 67.90 SD vs. 295 µm ± 69.04 SD), while the foveal central macular thickness (FCMT) was greater (270.91 μm ± 74.04 SD vs. 221 µm ± 37.25 SD). Conclusions: This research highlights the high incidence of macular complications. The results of our study indicate the importance of regular monitoring of RP patients and early intervention to avoid further complications in this group of subjects with severe visual field impairment to avoid further central vision loss.

## 1. Introduction

The definition of retinitis pigmentosa (RP) includes a heterogeneous group of inherited disorders involving photoreceptor degeneration, leading to visual loss with hemeralopia. RP is a major cause of visual disability and blindness, affecting more than 1.5 million patients worldwide. It is the most common inherited retinal dystrophy, with a worldwide prevalence of approximately 1:4000, according to studies from America. However, reports vary from 1:9000 in the Korean population to as high as 1:750 in the Indian population [1,2,3]. The estimated prevalence in the European population ranges from 23,927 (0.03%) in Germany to 20,424 (0.03%) in France, 19,816 (0.03%) in the UK, 4799 (0.008%) in Italy, and 3617 (0.008%) in Spain, with non-syndromic RP accounting for 80% of all RP patients [2,3,4,5,6].

The initial symptoms of RP are difficulty with dark adaptation and reduced night vision, or hemeralopia, which is followed by a progressive loss of the visual field (VF) in a concentric pattern due to the progressive degeneration of photoreceptors and retinal pigment epithelium (RPE). The appearance of the fundus is heterogeneous in RP. In the early stages, it may appear normal or may include a waxy-pale optic nerve head, attenuated retinal vessels, and pigment on the retina. Macular function is usually relatively well preserved until the later stages, although anatomical abnormalities in the central retina may appear early in the disease progression [7,8,9,10,11]. Vision acuity can be significantly affected by posterior subcapsular cataracts, occurring in approximately 45% of RP patients; cataract surgery can be performed in the presence of macular involvement with due precaution [12]. The etiopathogenesis responsible for cataracts is unknown, although a possible association with chronic inflammation was recently suggested [13,14].

The severity of RP can be evaluated using different examination and imaging modalities, including visual acuity (VA), VF testing, electroretinography (ERG), spectral domain optical coherence tomography (SD-OCT), and fundus autofluorescence. SD-OCT is a non-invasive imaging technique that measures the visualization, morphology, reflectivity, and quantification of different retinal layers; it represents a useful tool in the diagnosis and follow-up of possible changes. SD-OCT imaging may also be useful for detecting other macular abnormalities in RP patients [11,12,13,14,15], such as cystoid macular edema (CME), the most common finding, epiretinal membrane (ERM) formation, vitreomacular traction (VMT), lamellar macular holes (LMHs), and full-thickness macular holes (FTMHs) [11,16,17]. CME has been reported in up to 50% of patients with RP [11]. The shortening of photoreceptor outer segments is the earliest histopathological finding in RP, which, on SD-OCT imaging, corresponds to the disorganization of the outer retinal layers. This initially affects the interdigitation zone, followed by the ellipsoid zone (EZ), and eventually the external limiting membrane [17,18,19,20]. In the intermediate stages of RP, the thinning of the outer segments and a reduction in ONLT are observed. In the late stages of the disease, there is a complete loss of both the outer segments and the ONL (outer nuclear layer) [21]. Generally, the inner nuclear layer and the ganglion cell layer are relatively well preserved in severe RP cases. On the contrary, many studies have shown retinal thinning in RP patients due to the reduction in the thickness of the external layers [22,23].

As is widely acknowledged, several studies have revealed a correlation between the visual acuity of RP patients and the condition of the EZ line [24,25,26,27]. Furthermore, it has been shown that the width of the EZ line is also associated with a decrease in VF sensitivity, and a linear correlation between decreased VF and the thinning of the outer segments has been described [17].

The aim of this study was to investigate the prevalence of macular complications detected using SD-OCT in a large Caucasian cohort of RP patients and to highlight the major alterations in chorioretinal structure. Furthermore, we wanted to compare the macular structural changes in eyes affected by RP with those in healthy controls.

## 2. Materials and Methods

A total of 307 patients with clinical and genetic diagnoses of RP were enrolled in this retrospective, observational, and descriptive study. They were followed up for over 10 years, from 2010 to 2021, at the Center for Rare Retinal Diseases of the Umberto I Policlinic, Sapienza University of Rome, Italy. The research followed the tenets of the Declaration of Helsinki, and informed consent was given by all subjects of the study, 307 RP patients and 79 healthy controls. The protocol was reviewed and approved by the Ethics Committee of the Sapienza University of Rome (Rif. 15/2020RP DU Sense Organs, Ethics Committee Rif. 6502 Prot. 0920/2021). AI tools have been used in the preparation of the manuscript. The language corrections were made based on chatGPT.

### 2.1. Patients and Clinical Evaluation

Patients with a well-defined diagnosis of RP were enrolled. The inclusion criteria were history of hemeralopia, evidence of peripheral VF constriction with Goldmann and/or Humphrey testing, characteristic findings on fundus examination of bone spicule-like pigment clumping and attenuation of the retinal vessels, reduced visual evoked potentials (VEP), and A and B waves of the full-field ERG unrecordable, performed according to the International Society for Clinical Electrophysiology of Vision (ISCEV) standard [18]. All subjects with incomplete or unclear medical records were excluded. Other exclusion criteria were uncertain diagnosis; syndromic RP, such as Usher, Refsum, or Bardet–Biedl syndrome; other inherited retinal dystrophy diagnosis; ocular fundus without pigment deposit; marked ocular opacities; macular atrophy; history of cataract and vitreoretinal surgery; previous intravitreal therapy; and/or intraocular surgery. Also, participants with myopia, hyperopia, or astigmatism with spherical equivalent ≥3 diopters; ocular vascular and inflammatory diseases; and cardiovascular, dysmetabolic, neurological, and cancer diseases were excluded. After the evaluations were carried out on the 307 enrolled patients, 89 were excluded from the study at the start of the assessment or during the tests, and 218 continued the investigations. The patients and healthy controls underwent comprehensive eye exams, including BCVA in logMAR measured using the Snellen chart at 4 m, slit lamp biomicroscopy, intraocular pressure measurement with Goldmann applanation tonometry, fundus examination performed with indirect ophthalmoscopy, and SD-OCT obtained using the Spectralis OCT (Spectralis_HRA/OCT Heidelberg Engineering, Heidelberg, Germany) (Figure 1).

### 2.2. SD-OCT Measurements

Using SD-OCT, a horizontal raster scan centered on the fovea (5.8 × 4.3 mm) and a linear horizontal ‘dense’ foveal scan that covered most of the macular region between the vascular arcades were obtained for both eyes. In the horizontal raster scan, retinal thickness and volume were divided into nine ETDRS areas consisting of a central spot representing the FCMT, or fovea centralis, in an inner circular zone with a 1 mm diameter, as well as two outer rings with a 3 mm diameter divided into four quadrants for each retinal section: superior, inferior, temporal, and nasal. Thickness and volume for each section were noted (Figure 2).

The central measures of the EZ, ONLT, and SCT were manually obtained on a linear horizontal or vertical foveal scan. The EZ was defined as the line between the nasal and temporal boundaries where this layer drops to the outer segment of photoreceptors. The ONLT was measured as the most hypo-reflective external layer at the central fovea. The SCT scan (Enhanced Depth Imaging Mode, or EDI) was obtained by quantifying the subfoveal vertical distance between Bruch’s membrane interface and the sclerochoroidal junction, including measurements of the Sattler and Haller layers at high resolution (Figure 3).

Choroidal layer thickness was measured in three locations beneath the fovea: centrally and at two lateral points (left and right, within 1500 µm). The mean of these measurements was used in the analyses. Additionally, OCT images were reviewed to confirm the presence of CME, ERM, and other macular complications. CME was identified by the presence of cystic spaces with reflective walls on two or more consecutive macular raster scans. The ERM was defined as a hyper-reflective membrane adjacent to the internal limiting membrane, with or without anatomical distortion of the fovea. Macular complications were classified based on the International Vitreomacular Traction Study (IVTS) [19] definitions:

(a)VMT: presence of perifoveal vitreous cortex detachment from the retinal surface, with macular attachment of the vitreous cortex within a 3 mm radius of the fovea associated with distortion of the foveal surface, intraretinal structural changes, and/or elevation of the fovea above the RPE.(b)FTMH with VMT: a full-thickness foveal lesion that interrupts all macular layers.(c)The holes were classified by size as small holes (<250 μm), medium (250–400 μm), or large (>400 μm).(d)Small, medium, or large FTMH without VMT.(e)Lamellar macular hole (LMH): characterized by an irregular foveal contour, inner retina defect, intraretinal splitting (schisis), and an intact photoreceptor layer.

### 2.3. Statistical Analysis

Descriptive statistics for the cohort, stratified into research and control groups, were presented as means and standard deviations (SD) or medians and interquartile ranges for continuous variables; frequencies and percentages were reported for categorical variables. Fisher’s exact test was used for the categorical variable comparisons. For the comparison of continuous variables between two independent groups, two-tailed Student’s *t*-test or Mann–Whitney test was employed. Statistical significance (*p*), odds ratio (OR), and 95% confidence intervals (CI) were reported. Differences were considered statistically significant at *p* values < 0.05. Data were analyzed using the Statistical Package for the Social Sciences (SPSS), version 27 (IBM corporation, Armonk, NY, USA).

## 3. Results

### 3.1. Population

A total of 218 patients (436 eyes) out of the 307 patients initially enrolled were included for further morphological investigations of the macula. Of these, 99 were male (45.41%) and 119 were female (54.59%), with a mean age of 52.17 years (±17.25 SD) (Table 1). The control group consisted of 79 individuals, of whom 45 (56.96%) were male and 34 (43.03%) were female, with a mean age of 61.02 years (±5.31 SD) (Table 1).

The test results were normal in the control group and showed varyind degrees of abnormality in the RP group. No significant differences in sex or age were observed among the study participants.

### 3.2. OCT Measurements

Table 2, presents the morphological data obtained by SD-OCT in the patient group. The mean foveal central macular thickness (FCMT), or fovea centralis, was 270.91 μm (±74.04 SD), and the mean central volume within 1 mm was 0.21 mm^3^ (±0.05 SD). The mean ellipsoid zone (EZ) length was 2741.18 μm (±1859.76 SD), the mean ONLT was 84.73 μm (±35.44 SD), and the mean subfoveal choroid thickness (SCT) was 193.03 μm (±67.90 SD) (Table 2).

A total of 103 eyes (23.62%) exhibited CME, with 37 (16.97% of all patients) showing bilateral involvement. VMT was observed in 123 eyes (28.21%), with bilateral involvement in 36 cases (16.51% of all patients). The ERM was identified in 199 eyes (45.75%), with 67 cases (30.73% of all patients) exhibiting bilateral involvement. Additionally, 12 cases (2.75%) of LMH were identified, 3 of which were bilateral (1.38% of all patients). No cases of FTMH were identified (Table 3).

### 3.3. A Variety of Macular Alterations in RP Patients

Among all patients studied, 39 (17.88%) exhibited two mixed complications in both eyes. In our cohort, 160 eyes (36.69%) presented with two mixed central foveal alterations simultaneously. In contrast, a single complication in both eyes was observed in 143 patients (65.59%). An association between VMT and ERM was found in 64 (14.68%) eyes, with 14 patients (6.42%) showing bilateral VMT and ERM. CME and VMT were reported in 30 (6.88%) eyes; in 7 patients (3.21%), CME and VMT were bilateral. CME and ERM were observed in 56 (12.84%) eyes in 14 patients (6.42%) presenting bilateral CME and ERM. Of the 12 LMH cases detected, 4 (0.92%) were associated with VMT, all of which were bilateral; 6 (1.38%) were in combination with ERM, with 2 cases being bilateral (0.92%) (Table 3). Additionally, 18 eyes (4.13%) exhibited ERM, VMT, and CME simultaneously, with 3 cases (1.38%) being bilateral. LMH associated with both ERM and VMT was found in 4 eyes (0.92%), all of which were bilateral (Table 3). Only 43 patients (19.72%) and 137 eyes (31.42%) displayed none of the macular complications (Table 3) (Figure 4). the volume and thickness of the outer nuclear layer (ONLT),

### 3.4. Visual Acuity in RP Patients with Different Macular Alterations

Table 4 presents patient ages categorized by structural complications and the number of eyes (one or both). VA was reported according to complications using the logarithm of the minimum angle of resolution (logMAR). The largest group of patients suffering from vitreoretinal and retinal pathologies was between 41 and 70 years of age, with prevalent bilateral involvement. VA for the various pathologies examined primarily ranged between 0.0 and 0.2 logMAR (Table 4).

### 3.5. A Comparative Analysis of the Two Groups

The results of our study demonstrated a notable reduction in SCT among RP patients compared to healthy controls, with mean values of 193.03 µm (±67.90 SD) and 295 µm (±69.04 SD), respectively. Additionally, FCMT was markedly increased in RP patients, averaging 270.91 μm (±74.04 SD), whereas in healthy subjects, it was 221 (±37.25 SD). Furthermore, foveal thickness in the central 3 mm was significantly higher in the patient group at 300.60 µm (±47.71 SD) compared to 245 µm (±38.21 SD) in the control group. A comparison of mean values between the two groups is presented in Table 5 and Figure 5 and Figure 6.

## 4. Discussion

RP encompasses a group of inherited disorders characterized by photoreceptor degeneration and subsequent visual loss. Previous studies have shown that macular complications are frequently associated with RP. In this retrospective study, we investigated the structure and prevalence of macular complications, including CME, VMT, ERM, and LMH, in 218 patients with RP using SD-OCT.

The most crucial parameters for assessing disease progression and visual outcomes are morphological data. In a recent study, Poornachandra et al. reported that the mean horizontal extent of the preserved EZ was 2856.4 μm (range 518–8799 μm) [22]. The ONL has also been a focus of investigations, with researchers noting its early involvement in the intermediate stages of the disease. Sousa et al. and Sayo et al. observed a reduced mean subfoveal ONLT of 31.6 μm (±20.7 μm) and 44.5 μm (±16.4 μm), respectively, in RP patients compared to controls (66.5 μm ± 7 μm) [28,29,30].

Several studies have suggested that SCT may also be involved in this process [31,32]. Iovino et al. reported a significantly increased SCT in the RP group with CME compared to patients without CME (294.2 ± 110.9 μm vs. 198.1 ± 75.5 μm, respectively; *p* < 0.001), indicating a potential correlation between SCT and CME pathogenesis [32]. Studies by Guo et al. have also suggested characteristics of the pachychoroidal spectrum in both normal subjects and individuals with various pathologies, including 11 patients with RP [23,27,31]. Although the sample size in the study is acknowledged to be limited, further investigation into the two classification types identified by Guo et al. [23] is recommended.

In the present study, the mean SCT was found to be 193.03 µm (±67.90 µm), a value that is significantly reduced compared to that observed in the control group, which exhibited a mean SCT of 295 µm (±69.04 µm), as measured using SD-OCT. Similarly, the normal foveal thickness in the central 3 mm was 245 µm (±38.21 µm), while in our patients, it was significantly higher, 300.60 µm (±47.71 µm). This may be attributed to biochemical alterations, the etiology of which remains uncertain and which are responsible for morphological variations due to the presence of CME, VMT, and ERM in 23.62%, 28.21%, and 45.75% of the RP eyes examined, respectively.

We propose that the finding of CME in 23.62% of our selected patients, all with well-defined macular alterations, may be linked to altered choroidal circulation and metabolism. This suggests that it may be part of a pachychoroid spectrum. It is widely acknowledged that retinal function, particularly that of the photoreceptors, is inextricably linked to the health of the choroidal vasculature, which enables the transfer of metabolites and oxygen to the outer retina via the RPE.

In fact, several authors have presented evidence indicating that the OCT examination reveals a globally altered choroidal structure and reduced choroidal vascularization in RP pathologies compared to control subjects [27,33,34,35]. Pachychoroid can be considered a clinical entity characterized by a reduction in the thickness of the choriocapillaris in the presence of dilated and hyperpermeable choroidal vessels. This entity encompasses chorioretinal interface disorders that are associated with progressive dysfunction of the retinal pigment epithelium (RPE). A variety of ischemic, inflammatory, and genetic factors at the retinochoroidal interface have the potential to result in abnormalities in the tissue layers, which may subsequently lead to macular disorders and further visual impairment in patients with RP.

The analysis of the results has led to the expansion of the structural data pertaining to RP patients, and it has been established that the most prevalent retinal disorders are CME, ERM, and VMT.

To date, the etiology of CME remains unknown. However, Strong et al. recently proposed several mechanisms that may contribute to its formation. These include the breakdown of the blood–retina barrier, impaired function of the RPE pumping mechanism, Müller cell edema and dysfunction, anti-retinal antibodies, and VMT [11]. It has been proposed that in up to 36% of patients with RP, the formation of ERM may result from idiopathic pre-retinal glial cell proliferation or be secondary to an inflammatory process [10,36,37,38,39,40].

A great difference in the prevalence of CME among different studies has been reported, ranging from 8% in a cohort of 323 Japanese patients to 38% in a cohort of 124 patients of different ethnic backgrounds [41,42]. Liew et al. found CME in 58.6% of 169 patients, while Testa et al., studying 237 eyes at an Italian eye clinic, noted a CME prevalence of 20.4% [39,43,44]. Our results are similar to those obtained in Italian study groups.

The prevalence of vitreoretinal interface abnormalities associated with RP also varies across studies, with ERM and VMT rates ranging from 0.6% to 80.5% and from 0.8% to 13.6%, respectively [39,41,43,44]. Several authors reported a 0.5% prevalence of FTMH [39,41,43], while LMH was found in 5.8% of eyes in a study of 12 patients (12/145 = 8.2%) conducted by Fragiotta et al. [44].

### Study Limitations and Future Directions

The presence of significant tissue alterations, including extreme retinal thinning and the absence of identifiable retinal layers, made the assessment of OCT measurements challenging. The primary limitation of this study was its retrospective design; however, the findings regarding prevalence are particularly noteworthy. Furthermore, a comprehensive genetic panel was not available for all patients, and in some cases, negative genetic results were observed, precluding any assessment of the correlation between specific mutations and macular complications. Additionally, the single-center design of this study represents another potential limitation to the generalizability of our findings.

To date, only a limited number of studies have focused on the stromal and vascular characteristics of the choroidal tissue in the context of hereditary chorioretinal pathologies [45]. The most recent research on stem cell and chorioretinal autotransplantation in RP patients has already been published by our team as part of a larger research project that we intend to continue investigating [46,47]. Lastly, we plan to investigate the relationship between structural defects and genetic mutations in patients, dividing the RP cohort into groups based on disease stage to facilitate the development of an OCT-based staging system in conjunction with ocular electrophysiology assessments and psychophysical tests.

## 5. Conclusions

RP is a degenerative inherited retinal dystrophy classified as an incurable genetic disease. Our research highlighted the high incidence of macular alterations in a cohort of RP patients. OCT imaging can detect early morphological changes in the inner and outer retinal and choroidal layers, allowing for the monitoring of disease progression at various stages to help prevent further central vision loss.

In summary, we propose that a more comprehensive scientific approach is necessary to elucidate the etiology of tapetoretinal diseases, given the close correlation between the retina and choroidal health.

## Figures and Tables

**Figure 1 diagnostics-14-02409-f001:**
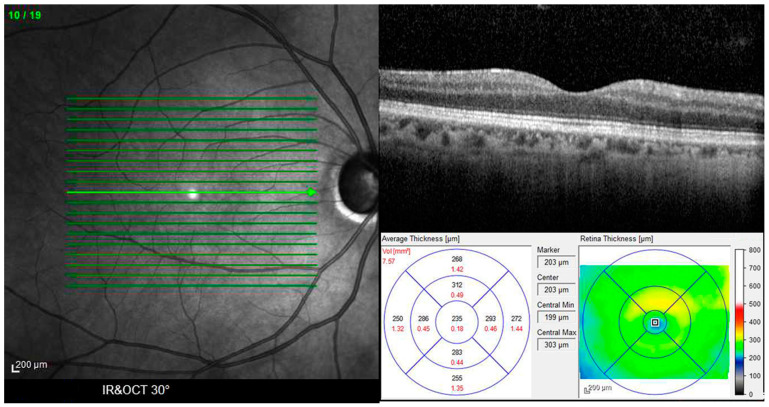
Infrared (IR) (**left panel**) and optical coherence tomography (SD-OCT) (**right panel**) fundus image of a normal subject. A horizontal raster scan centered on the fovea (5.8 × 4.3 mm) and a linear horizontal ‘dense’ foveal scan of the macular region between the vascular arcades were obtained for both eyes. On horizontal raster scan, retinal thickness and volume were divided into nine Early Treatment Diabetic Retinopathy Study (ETDRS) areas, consisting of a central spot representing the fovea centralis (FCMT), an inner circular zone with a 1 mm diameter, and two outer rings with a 3 mm diameter, divided into four quadrants for each retinal section, superior, inferior, temporal, and nasal (**bottom panel**).

**Figure 2 diagnostics-14-02409-f002:**
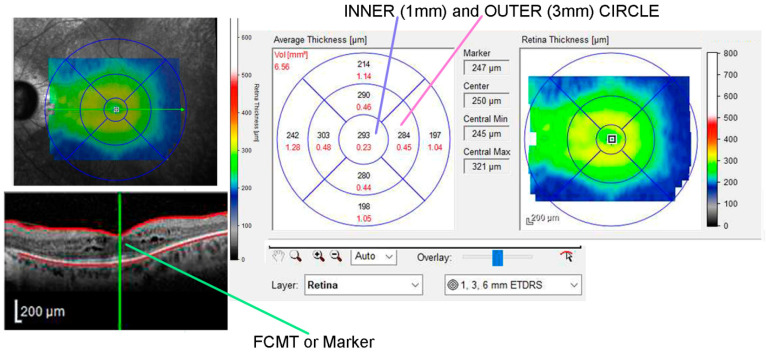
Spectral domain-optical coherence tomography (SD-OCT) of a retinitis pigmentosa (RP) patient affected by cystoid macular edema (CME). In the left panel, the green line indicates the thickness of the central macular fovea (FCMT), or fovea centralis. In the right panel, the central 1 mm circle and the outer 3 mm circle are highlighted, where measurements were taken for the RP and control groups.

**Figure 3 diagnostics-14-02409-f003:**
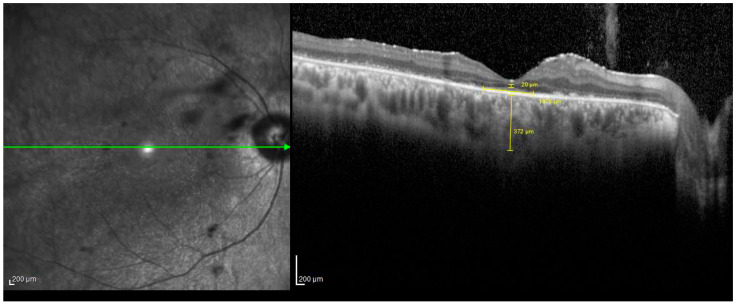
Spectral domain-optical coherence tomography (SD-OCT) of a patient affected by retinitis pigmentosa (RP). In the right panel, the yellow line indicates the subfoveal choroid thickness (SCT) in the EDI SD-OCT scan (Enhanced Depth Imaging Mode) measured by quantifying the subfoveal vertical distance between Bruch’s membrane interface and the sclerochoroidal junction, including measurements of the Sattler and Haller layers at a high resolution (372 µm).

**Figure 4 diagnostics-14-02409-f004:**
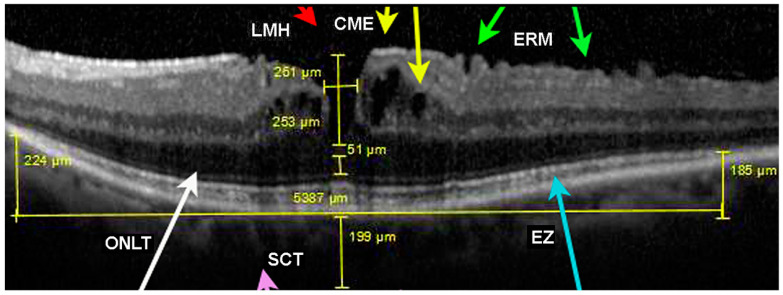
SD-OCT macular scan of a patient with RP, displaying a lamellar macular hole (LMH); (red arrow), cystoid macular edema (CME) with micro- and macro-pseudocysts (yellow arrows); an epiretinal membrane (ERM) (green arrows); outer nuclear layer thickness (ONLT) (white arrow); subfoveal choroid thickness (SCT) (pink arrow) and an ellipsoid zone (EZ) (blue arrow) that ends at the edges of the macular region, characteristic of hereditary retinal dystrophy.

**Figure 5 diagnostics-14-02409-f005:**
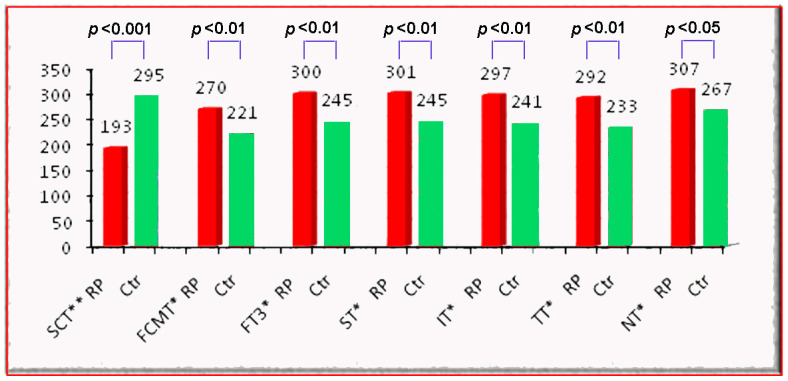
Comparison of thickness measurements between the retinitis pigmentosa (RP) patient group and control group (Ctr). Values are presented as group averages. Measurements include subfoveal choroidal thickness (SCT) within 1.5 mm (in μm); foveal thickness at the central point (FCMT) (μm); foveal thickness in the 3 mm outer circle (FT3) (μm); and superior (ST), inferior (IT), temporal (TT), and nasal thickness (NT) in the 3 mm outer circle (μm). Statistically significant differences are marked by asterisks (*) with *p* value < 0.05.

**Figure 6 diagnostics-14-02409-f006:**
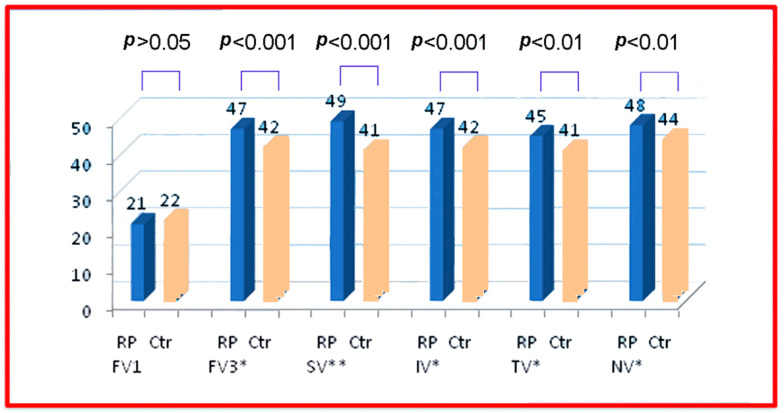
Comparison of the volume measurements between the retinitis pigmentosa (RP) patient group and control group (Ctr). Values are presented as group averages. Measurements include foveal volume in the 1 mm inner circle (FV1) (mm^3^); foveal volume in the 3 mm outer circle (FV3) (mm^3^); and superior (SV), inferior (IV), temporal (TV), and nasal volume (NV) in the 3 mm outer circle (mm^3^). Statistically significant differences are marked by asterisks (*) with *p* value < 0.05.

**Table 1 diagnostics-14-02409-t001:** Demographic characteristics of the study cohort evaluated using spectral domain optical coherence tomography (SD-OCT).

Cohort Data	Patients	Controls
Total	218	79
Eye	436	158
Mean age (±SD)	52.17 (±17.25)	61.02 (±5.31)
Male (%)	99 (45.41%)	45 (56.96%)
Female (%)	119 (54.59%)	34 (43.03%)

**Table 2 diagnostics-14-02409-t002:** Patient characteristics based on pectral domain optical coherence tomography (SD-OCT) examination.

**ONLT (Mean ± SD) ** **84.73 µm ± 35.44**	**Macular Sector**	**Mean Thickness** **in µm ± SD**	**Mean Volume** **in mm^3^ ± SD**
	Macular fovea (1 mm inner circle)	270.91 ± 74.04	0.21 ± 0.05
**SCT (mean ± SD)** **193.03 µm ± 67.90**	Fovea (3 mm inner circle)	300.60 ± 47.71	0.47 ± 0.07
	Superior (3 mm outer circle)	301. 28 ± 44.38	0.49 ± 0.13
**EZ (mean ± SD)** **2741.18 µm ± 1859.76**	Inferior (3 mm outer circle)	297.59 ± 51.12	0.47 ± 0.08
	Temporal (3 mm outer circle)	292.02 ± 49.30	0.45 ± 0.07
	Nasal (3 mm outer circle)	307.11 ± 44.69	0.48 ± 0.07

Foveal central macular thickness: FCMT; Outer nuclear layer thickness: ONLT; Standard deviation: SD; subfoveal choroidal thickness: SCT; length of the ellipsoid zone: EZ.

**Table 3 diagnostics-14-02409-t003:** Prevalence of macular complications and their associations in 436 eyes from 218 patients with RP.

Number and Counts (%)		Number and Counts (%)	
**No complications**	137/436 eyes (31.42)	VMT + ERM	64/436 eyes (14.68)
**No complications**	43/218 pts (19.72)	Bilateral VMT + ERM	14/218 pts (6.42)
**One bilateral complication**	143/218 pts (65.59)	CME + VMT	30/436 eyes (6.88)
**Two mixed complications**	160/436 eyes (36.69)	Bilateral CME + VMT	7/218 pts (3.21)
**Two mixed complications**	39/218 pts (17.88)	CME + ERM	56/436 eyes (12.84)
**Three complications**	22/436 eyes (5.05)	Bilateral CME + ERM	14/218 pts (6.42)
**CME**	103/436 eyes (23.62)	LMH + VMT	4/436 eyes (0.92)
**Bilateral CME**	37/218 pts (16.97)	Bilateral LMH + VMT	2/218 pts (0.92)
**VMT**	123/436 eyes (28.21)	LMH + ERM	6/436 eyes (1.38)
**Bilateral VMT**	36/218 pts (16.51)	Bilateral LMH + ERM	2/218 pts (0.92)
**ERM**	199/436 eyes (45.75)	ERM + VNT + CME	18/436 eyes (4.13)
**Bilateral ERM**	67/218 pts (30.73)	Bilateral ERM + VMT + CME	3/218 pts (1.38)
**LMH**	12/436 eyes (2.75)	LMH + ERM + VMT	4/436 eyes (0.92)
**Bilateral LMH**	3/218 pts (1.38)	Bilateral LMH + ERM + VMT	2/218 pts (0.92)

Patients: pts; cystoid macular edema: CME; vitreomacular traction: VMT; epiretinal membrane: ERM; lamellar macular hole LMH; percentage: %.

**Table 4 diagnostics-14-02409-t004:** Patient age, best corrected visual acuity (BCVA), and number of affected patients categorized by type of complication.

AGE Years/pts	ERM pts (Both Eyes)	CME (Both Eyes)	VMT PTS (Both Eyes)	LMH pts (Both Eyes)
<20/15	14 (5)	5 (2)	8 (3)	1 (0)
21–30/16	21 (7)	8 (2)	10 (4)	2 (1)
31–40/16	16 (5)	10 (4)	6 (2)	0 (0)
41–50/41	38 (11)	20 (7)	21 (4)	1 (0)
51–60/58	43 (14)	25 (9)	37 (14)	1 (0)
61–70/39	34 (14)	15 (5)	23 (5)	5 (1)
71–80/27	28 (8)	17 (7)	15 (2)	2 (1)
81–90/6	5 (2)	3 (1)	2 (0)	0 (0)
**BCVA**	**ERM**	**CME**	**VMT**	**LMH**
<+1	9	6	11	/
+1 or +0.9	13	4	4	/
+0.8 or +0.7	15	9	6	/
+0.6 or +0.5	15	15	7	5
+0.4	8	6	4	1
+0.3	6	7	7	3
+0.2	28	10	18	3
+0.1	11	7	6	1
0.0	29	15	24	1

Best corrected visual acuity (BCVA) in the logarithm of the minimum angle of resolution (logMAR); patients: pts; epiretinal membrane: ERM; cystoid macular edema: CME; vitreomacular traction: VMT; lamellar macular hole: LMH.

**Table 5 diagnostics-14-02409-t005:** Comparison of the variables between patient and control groups.

Means of the Variables	Patients	Controls	*p* Value
SCT (within 1.5 mm)	193.03 μm ± 67.90 SD	295 μm ± 69.04 SD	<0.001
FCMT	270.91 μm ± 74.04 SD	221 μm ± 37.25 SD	<0.01
Foveal volume (inner circle of 1 mm)	0.21 mm^3^ ± 0.05 SD	0.22 mm^3^ ± 0.11 SD	>0.05
Foveal thickness (outer circle of 3 mm)	300.60 μm ± 47.71 SD	245 μm ± 38.21 SD	<0.01
Foveal volume (outer circle of 3 mm)	0.47 mm^3^ ± 0.07 SD	0.42 mm^3^ ± 0.15 SD	<0.001
Superior thickness (outer circle of 3 mm)	301.28 μm ± 44.38	245.13 μm ± 19.01	<0.01
Superior volume (outer circle of 3 mm)	0.49 mm^3^ ± 0.13	0.41 mm^3^ ± 0.4	<0.001
Inferior thickness (outer circle of 3 mm)	297.59 μm ± 51.12	241.34 μm ± 14.52	<0.01
Inferior volume (outer circle of 3 mm)	0.47 mm^3^ ± 0.08	0.42 mm^3^ ± 0.85	<0.001
Temporal thickness (outer circle of 3 mm)	292.02 μm ± 49.30	233.41 μm ± 18.23	<0.01
Temporal volume (outer circle of 3 mm)	0.45 mm^3^ ± 0.07	0.41 mm^3^ ± 0.91	<0.01
Nasal thickness (outer circle of 3 mm)	307.11 μm ± 44.69	267.32 μm ± 20	<0.05
Nasal volume (outer circle of 3 mm)	0.48 mm^3^ ± 0.07	0.44 mm^3^ ± 0.81	<0.01

Foveal central macular thickness: FCMT; subfoveal choroidal thickness: SCT; standard deviation: SD.

## Data Availability

The data that support the findings of this study can be made available upon request from the corresponding author, M.N., under certain conditions. The data are not publicly available due to their containing information that could compromise the privacy of research participants.

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
