# Peer review of "Macular Alterations in a Cohort of Caucasian Patients Affected by Retinitis Pigmentosa"

_diagnostics, 2024, doi:10.3390/diagnostics14212409_

Round 1

Reviewer 1 Report

Comments and Suggestions for Authors

The authors state they want to study the prevalence of Macular structural changes in this large Caucasian Cohort of RP patients and compare to healthy normals. They provide a well written manuscript with a good biography. The introduction is a bit long , there is an entire paragraph about the incidence of various genes . Yet  the results  described in this paper do not  include genetic diagnoses ,  so I suggest to shorten it significantly. 

Material and methods: Intra ocular surgery: Did you exclude all the RP patients who underwent cataract surgery? Please explain & correct "tumor diseases", do you mean cancer diagnosis?   Patients with RP can have astrocytic hamartoma, were those excluded? Why genetic counseling is mentioned when there was no mentioning of family history, acquisition of pedigree, genetic testing and when the paper does not have specific genetic categories . Refraction and detailed assessment of refractive error and axial length would be very important if one is to use OCT measurements for comparative and quantitative measurements, might be more important than Goldmann applanation tonometry .  Furthermore refraction is  be important when comparing results of this large Caucasian Cohort of RP patients with other ethnic groups where myopia is a lot more common.     Did you include patients with Choroideremia? It appears there is a higher incidence of macular holes in choroideremia . 

A picture of EDI OCT with indication of the SCT would be very helpful.

line 174 : CME , non-reflective cystOID space (b/o there is no wall around the cyst it is not cystic)

Table 3  complications do you mean associated vitreo-macular interface disorders ? 

Discussion: line 417 what are dioptric media opacities?

Comments on the Quality of English Language

Some editing by a native English speaker is advisable.

Author Response

REVIEWER 1

The authors state they want to study the prevalence of Macular structural changes in this large Caucasian Cohort of RP patients and compare to healthy normals. They provide a well written manuscript with a good biography. The introduction is a bit long , there is an entire paragraph about the incidence of various genes . Yet  the results  described in this paper do not  include genetic diagnoses ,  so I suggest to shorten it significantly. 

  • We thank the Reviewer for the positive comments. The Introduction section has been revised and reduced. The paragraph on genetics has been eliminated.

Material and methods: Intra ocular surgery: Did you exclude all the RP patients who underwent cataract surgery?

  • We excluded patients who had undergone surgery and reported this in the inclusion and exclusion criteria.

Please explain & correct "tumor diseases", do you mean cancer diagnosis?   

  • We corrected it.

Patients with RP can have astrocytic hamartoma, were those excluded?

  • Patients affected by other pathologies were excluded.

Why genetic counseling is mentioned when there was no mentioning of family history, acquisition of pedigree, genetic testing and when the paper does not have specific genetic categories.

  • The Reviewer makes a good point. We have eliminated unnecessary sentences.

Refraction and detailed assessment of refractive error and axial length would be very important if one is to use OCT measurements for comparative and quantitative measurements, might be more important than Goldmann applanation tonometry.  Furthermore refraction is  be important when comparing results of this large Caucasian Cohort of RP patients with other ethnic groups where myopia is a lot more common.     

  • The patients examined had a refractive defect of no more than 3 negative or positive diopters including the spherical equivalent for astigmatism (see Methods section). This is to avoid errors of evaluation.

Did you include patients with Choroideremia? It appears there is a higher incidence of macular holes in choroideremia.

  • No, these patients were excluded.

A picture of EDI OCT with indication of the SCT would be very helpful.

  • Great advice, figure 4 highlights the choroid (EDI).

line 174 : CME , non-reflective cystOID space (b/o there is no wall around the cyst it is not cystic)

  • We changed the sentence in line 174.

Table 3  complications do you mean associated vitreo-macular interface disorders ? 

  • Yes, that's right.

Discussion: line 417 what are dioptric media opacities?

  • We changed the word to "cataract".

Comments on the Quality of English Language. Some editing by a native English speaker is advisable.

  • Native English speaker revised the manuscript.

Reviewer 2 Report

Comments and Suggestions for Authors

Authors have described the results of a clinical study of o investigate the prevalence of macular disorders detected by SD-OCT in a large Caucasian cohort of RP patients highlighting the major alterations in chorioretinal structure. They also performed a comparison of macular structural changes in eyes affected by retinal dystrophies with those in healthy controls. This is a nice  with high clinical values especially for Caucasian. I urge authors to incorporate the following comments to enhance the quality and visibility of the paper.

1. Authors need to provide the reference for Fig 1 unless they used the image from their datasets. It also looks strange to have a white gradient border rather than sharp border for the image. Could authors explain why ? Is this 

2. Can authors provide p value in Fig 3 and 4.

3. Can authors provide the retina of RP patient (not Fig 1) and healthy control side by side to have a better understanding for the readers.

4. How did authors measured the thickness ? Is it a custom built retinal layer segmentation program or commercially available one. Pls specifically mention about this tool in the text.

5. Since authors used OCT as a tool for longitudnal study of the retina, it is advised to add  a couple of sentences (with suggested reference) about the importance of OCT in ocular imaging of human and experimental animals. Authors may add " OCT is a non-invasive optical imaging modality that allows the in vivo visulaization of retinal and choroid in clinical ophthalmology and in preclinical investigations[1-2]. 

(1) https://doi.org/10.1016/j.omtn.2022.04.015

(2) https://doi.org/10.3390/diagnostics14070764

Author Response

REVIEWER 2

Authors have described the results of a clinical study of o investigate the prevalence of macular disorders detected by SD-OCT in a large Caucasian cohort of RP patients highlighting the major alterations in chorioretinal structure. They also performed a comparison of macular structural changes in eyes affected by retinal dystrophies with those in healthy controls. This is a nice  with high clinical values especially for Caucasian. I urge authors to incorporate the following comments to enhance the quality and visibility of the paper.

  • We thank the Reviewer for the positive comments.

  1. Authors need to provide the reference for Fig 1 unless they used the image from their datasets. It also looks strange to have a white gradient border rather than sharp border for the image. Could authors explain why ? Is this
  • All figures in the manuscript are from our patients. Figure 1 which in the Revision became number 2 had a blurred border to give a different look. We have reported the original figure.

  1. Can authors provide p value in Fig 3 and 4.
  • The p-value has been added to figures 3 and 4 (Rev. 5 and 6).

  1. Can authors provide the retina of RP patient (not Fig 1) and healthy control side by side to have a better understanding for the readers.
  • We have inserted figure 1, SD-OCT examination of a normal subject. We did not place the figures side by side because the details would not have been visible.

  1. How did authors measured the thickness? Is it a custom built retinal layer segmentation program or commercially available one. Pls specifically mention about this tool in the text.
  • The instrument we used, SD-OCT, has a program that allows detailed measurement of the structures, and in the methods section, we have described the precise start and end points of the measurement in detail.

  1. Since authors used OCT as a tool for longitudnal study of the retina, it is advised to add a couple of sentences (with suggested reference) about the importance of OCT in ocular imaging of human and experimental animals. Authors may add " OCT is a non-invasive optical imaging

modality that allows the in vivo visulaization of retinal and choroid in clinical ophthalmology and in preclinical investigations[1-2].

(1) https://doi.org/10.1016/j.omtn.2022.04.015

(2) https://doi.org/10.3390/diagnostics14070764

  • The sentence and the 2 references have been inserted into the manuscript (see page 15).

Reviewer 3 Report

Comments and Suggestions for Authors

see included file

Comments on the Quality of English Language

I would suggest help with lingustic editing by professioanl English specialist. Some phrases are difficult to understand and the main reason is the structure of the sentences, order of the words in the sentence and phrasing. The spelling errors are minor. 

Author Response

REVIEWER 3

The study by Marcella Nebbioso et al. explored retinal morphology using optical coherence

tomography (OCT) in a cohort of 307 patients affected with retinitis pigmentosa (RP). This is a singlecenter retrospective cross-sectional study, which focused primarily on three distinct OCT parameters, such as ellipsoid zone volume, thickness of the outer nuclear layer and subfoveal choroidal thickness. The authors described alterations in these parameters in studied population, as well as distribution of macular complications affecting RP patients, such as cystoid macular oedema, epiretinal membranes and vitreomacular traction. While I appreciate the effort of the authors and recognise the importance of studying retinal morphology in patients with RP, the study and the way of presenting and discussing data would need significant improvement.

I will address my general comments below, followed by more specific recommendations for

improvement. Broad comments.

  • We thank Reviewer 3 who allowed us to make the requested changes to improve the study's drafting.

1) The title of this manuscript could be more specific, highlighting main findings.

  • We changed the title of the manuscript: “Morphological findings with oct and related complications in a cohort caucasian of retinitis pigmentosa.”

2) The objective of this study is not entirely clear to me. As it appears in the abstract: 1st

objective is a review of current knowledge, whereas the 2nd is a research study. I would

suggest to be more precise. The conclusions are very diffuse in the abstract, need to be better defined.

  • The purpose and conclusions of the study have been implemented compatible with the number of words that the Editorial Office allows us to report in the description of the Abstract.

3) The Introduction Section is very general about RP, and many aspects of disease pathology.

There is very little information on morphological studies, such as OCT, which is the main

focus of this manuscript. I recommend to shorten general introduction of RP, and focus a bit

on OCT as a technology, and OCT findings in RP patients that had been already published.

  • We have shortened and modified the Introduction section, as requested by other Reviewers, and added some considerations on the SD-OCT exam. The Discussion section includes other information on published studies compared with our works.

4) In the Material and methods, the authors describe other parameters used to study RP

patients, such as fundus imaging, VF and ERG. These findings, however, are not described in

the Results section or discussion in correlation to OCT findings, neither any conclusions are

drawn. The control group is slightly too small. This section is very long, would need to be

shorten with focus on main objective of this study.

  • In the Materials and Methods section, we have specified the aim of the study. The various tests were performed only to ascertain the diagnosis and the conditions of our patients, therefore to be certain of the inclusion and exclusion criteria; the tests did not concern the aim of the study. We have described only those procedures that involved data collected by SD-OCT. The statistical analysis was performed by a full professor of biostatistics, who ensured the validity of the analysis and the number of controls analyzed.

5) In the Results section, to me the main findings of this manuscript are: 1) the prevalence of

the macular complications in RP patients studied in this specific population, such as CME,

VMT and ERM; 2) The foveal volume and thickness. These two aspects should dominate in

the results section, and I would suggest that the correlation with other data is also described

(demographics, age, genotype, functional studies: VF and ERG). In general, I would suggest

adding subtitles in this section to increase readability.

  • We have included most of the data obtained in the Results section. Correlational studies require a separate manuscript; we are waiting for the genetic data to continue our study and include the other information (VF, ERG, etc.) suggested by the Reviewer.

6) The Discussion section need to be substantially edited and re-structured. This sections is too

general and main findings of this manuscript are not clear. It appears to me almost as a

review with a lot of data discussed and out of focus. Few paragraphs could be moved to

introduction section (first 3-4 paragraphs). I would recommend subtitles here describing

main topics discussed and correlation, also correlation between this study and what was

already published.

  • We have revised, modified, and restructured the Discussion section. We have also moved the first few paragraphs to the Introduction section.

7) In general, the manuscript is a bit out of focus, and I think this is the main limitation. The

authors should pay attention to main findings of this study, re-design study in order to try to

find more correlations (as mentioned in point (5), draw precise conclusions free of

speculations, and discuss the results accordingly.

  • This note from the Reviewer is certainly useful. As we have already reported in point 5, further data and correlations will be part of another manuscript, so as not to overload the concepts and add too many tables already transcribed in this paper. In the Discussion section, we have emphasised the importance of the choroid and the close morphological and functional dependence of the relationship between the outer retina and the choroid.

Specific comments

Abstract:

- row 17-19: reformulate and specify your objectives more clearly.

-We clarified

- row 29: correct to “mentioned above”

  • We corrected it.
  •  

- row 32-34: reformulate your conclusion, be more precise.

  • We have reformulated the conclusions.

Introduction:

In general, the introduction appears unstructured in terms of the content and writing style. Please,

introduce paragraphs to this section and start each paragraph with opening phrase. I would suggest shortening the text about general aspects of RP, and instead focus on OCT as a technique and OCT findings in RP since this is the most essential topic, which would give a good overview to your data.

  • Thanks for your comments and suggestions. We have reduced the section Introduction, removed some general introduction parts, and reformulated the approach to the OCT technique.

- Row 43: add few references

  • We added the references.

- Row 44: change “American studies” to “according to studies from America”

  • We corrected the words.

- Row 46: change “estimates of the prevalence” to “The estimated prevalence”

  • We corrected the words.

- Row 44: a statement starting with “RP can be classified” should start a new paragraph

  • We started the sentence with a new paragraph.

- Row 52: change phrasing and structure of this sentence

  • The sentence has been deleted at the request of Reviewer 1.

- Row 59: after “described” change to “yet several cases remain still unclassified”

  • We corrected the sentence.

- Row 60: start new paragraph with opening phrase “Concerning syndromic forms, Usher

syndrome…”

  • We corrected the sentence.

- Row 66: start new paragraph and change the structure of this statement starting with “The

initial symptoms of RP…”

  • We started the sentence with a new paragraph and changed the structure.

- Row 74: start this statement with phrase “the fundus appearance is heterogeneous in RP…”

and then describe features which may be present

  • We changed the sentence.
  •  

- Row 77-84: this paragraph may be not necessary in the context of this manuscript

  • The sentences have been eliminated.
  •  

- Row 84-98: shorten this paragraph and edit it in a way fitting better to the context of this

Manuscript

  • The phrases have been shortened and remodeled.

- Row 98-118: is the most essential part of the introduction. Please, edit the entire Introduction section accordingly, and as described in the Broad comments.

  • We changed the sentences.

Material and methods:

I would recommend introducing subtitles to this section, e.g. Patients; Clinical evaluation/ or Clinical data acquisition; OCT measurements; Statistics.

  • The requested subtitles have been introduced.

- Row 121: remove word “instrumental”

  • The word removed.

- Row 122: change “checked” to “followed up”

  • The word has been changed.

- Row 130: only patients with these fundus features were included to this study?

  • Yes, only patients with these characteristics were enrolled.

- Row 132: why VEP was taken into account in RP patients? Why only patients with

unrecordable ERG were included and not rod-cone dystrophy with mild/moderate/severe

reduced ERG?

  • Because we wanted to study a homogeneous group of patients with the same clinical and morpho-functional characteristics. The appropriate research and statistics are those carried out on well-selected individuals.

- Row 148-150: change the order of the words in this statement, start with “Using SD-OCT, a

horizontal raster scan cantered on the fovea…were obtained for both eyes…”

  • We changed the sentence.

- Row 179-190: please, shorten this section and be more precise.

  • We changed the sentences.

Results:

Please, structure this section for better readability, e.g. using subtitles emphasizing main findings. It is not clear now and main results of the study are not highlighted. Try to use active vs. passive voice, e.g. we found/ we showed/present.

  • We have inserted subtitles and used the active form.

- Row 210: rephrase a part of the statement “…to some extent, abnormal in RP group” and

write precisely what was abnormal

  • The responses of the various tests carried out on the patients altered.

- Row 215-217: this text should be shifted to Methodology section

  • The text was shifted.

- Row 217-223: summarize the results briefly and refer to Table 2, where they are already

presented

  • The request to modify has been executed.

- Row 238: rephrase this statement starting with “Of all studied patients (n=…), 39 (17.88%)…”

  • The request to modify has been executed.

- Row 248: correct the word “patients”

  • The word corrected.

- Row 245, the same in row 248: remove “instead” and use “whereas” here as the beginning of

a new sentence

  • The word was corrected.

- Row 250: finish this statement with phrasing (Table 4)

  • The request to modify has been executed.

- Row 263: please, restructure this sentence as e.g. “In our study, we found that RP patients

had significantly reduced SCT compared to healthy controls (numbers here)…. Furthermore,

FCMT was significantly higher

  • The request to modify has been executed.

- Row 269: change also the structure and phrasing of this sentence

  • The request to modify has been executed.

Discussion:

I would suggest organizing paragraphs using subtitles with main findings/highlights in this section to increase readability. This part of manuscript requires the most changes. It is not clear how the results of this manuscript contribute to the field, especially in the context of previously published data. Again, I would recommend to focus on morphological studies, specifically OCT in RP patients, and not to include too much general text about RP pathology.

- The first 3-4 paragraphs need to be extensively edited and maybe partly shifted to

Introduction section (rows 293-314)

- Row 329-331: the content is unclear, be more precise

- Row 332-336: please, rephrase this sentence. The content is speculative

  • Subtitles have been inserted and edits have been made as requested.

- Row 336-339: edit this sentence, please. Start with “It is commonly known that…”

  • The phrases have been remodeled.

- Row 347-361, the same in rows 378-394: the data already published are extensively

discussed and there is very little text how this data correlate to the results from this

manuscript. The same appears in many paragraphs of the Discussion section.

  • We have shortened and rephrased some sentences, and we think that the results of other researchers are useful for comparison.

- Row 363-365: please re-write and shorten this sentence

  • The phrases have been remodeled.

- Row 395: remove a word “Anyway” at the beginning of this statement. Again this paragraph

(row 395-407) about ocular inflammation describes the findings already published and does

not correlate clearly to the results of this manuscript

  • The paragraph has been shortened and reworded to explain the morphological changes in the patients.

- Row 408: please, remove this statement

  • The sentence has been removed.

- Row 409-414: this paragraph is not precise, and need to be edited

  • The paragraph has been modified.

- Row 418-421: please, rewrite and improve structure of this statement. This part of the

discussion section could also have a subtitle “Study limitations and future directions”

  • The requested changes have been made.

Conclusions:

- Row 438-440: change the part of the statement describing therapy to “gene and cell therapy,

and other interventional approaches”

- Row 440: change “little importance has been placed” to “there are only few studies which

focused on exploring….”

- Row 442: start this statement with “We believe that….”, then change “well-being of the

retina…” to “status of the retina and choroid”

  • The requested changes have been made.

Figures and tables:

Figure 1: enlarge the numbers on the OCT image if possible

  • 2 (in the previous manuscript Fig. 1) has been updated with a better visualization of the numbers.

Table 1: the number of controls is slightly too low

  • Useful note from the Reviewer and we will continue the study later. We followed the instructions of the full professor of biostatistics.
  •  

Table 4: I would recommend to present this table as the 1st one in the Results section and re-arrange the text in this section accordingly

  • Although the Reviewer made the right request, I would kindly ask you to accept the actual organization of the section. This change would imply substantial changes that we no longer can make, given the limited time. I thank you in advance for your possible understanding.

Figure 3, Figure 4: add different colors to columns representing patients and control group

  • The requested changes have been made.

Round 2

Reviewer 1 Report

Comments and Suggestions for Authors

There is a substantial improvement of the manuscript , the paper is much better organized with subtitles for statistics analysis and result section.

I hope the editors will be able to improve the quality of the written English language with some additional cosmetic  changes  

Figure 2: retinography?? To me it is a fundus picture that is upside down if it depicts a right eye. You show twice the same OCT  B scan image, please redraw only the lower section of the image to include all the arrows and annotations so you can delete the top right . 

cancer diseases are Malignancies 

Figure 6; interesting shades, why 2 different shades in the controls , does that mean anything? if not you may want to keep it simple 

Comments on the Quality of English Language

Yet some additional cosmetic changes are recommended, for instance conclusions of abstract section the sentence is awkward even in spoken english . population  "whose 99 were males" .... Morphological data considering all eyes 103 had CME "whose  

Author Response

Dear Prof. Slawomir Teper

Guest Editor Diagnostics Journal,

and

Chair Clinical Department of Ophthalmology,

Faculty of Medical Sciences in Zabrze, Medical University of Silesia in Katowice, Poland

[Diagnostics] Special Issue: Optical Coherence Tomography Imaging in Retinopathy: New Advances and Future Trends - Submission Deadline: 30 September 2024 (e-mail 7/08/2024).

New title recommended by Reviewer 3: “Macular alterations in the cohort of caucasian patients affected with retinitis pigmentosa”

       We thank you for allowing us to send the revised manuscript titled “Macular alterations in the cohort of caucasian patients affected with retinitis pigmentosa” by Marcella Nebbioso, Elvia Mastrogiuseppe, Eleonora Gnolfo, Marco Artico, Antonietta Moramarco, Fabiana Mallone, Samanta Taurone, Annarita Vestri, and Alessandro Lambiase.

We are grateful for the useful comments and valuable improvements to our paper. Those changes are highlighted within the manuscript. Responses to the Reviewer comments have been reported at the bottom of the letter.

We hope that your journal will consider the manuscript.    

Best regards

Marcella Nebbioso and Co-Authors

Marcella Nebbioso, MD, Prof. (orcid.org/0000-0002-5512-0849)

e-mail: marcella.nebbioso@uniroma1.it Phone: ++39/06/49975422 Fax: ++39/06/49975426

Department of Sense Organs, Sapienza University of Rome, piazz.le A. Moro 5, 00185 Rome,

Italy.

REVIEWER 1

Comments and Suggestions for Authors

There is a substantial improvement of the manuscript , the paper is much better organized with subtitles for statistics analysis and result section.

I hope the editors will be able to improve the quality of the written English language with some additional cosmetic  changes  

Figure 2: retinography?? To me it is a fundus picture that is upside down if it depicts a right eye. You show twice the same OCT  B scan image, please redraw only the lower section of the image to include all the arrows and annotations so you can delete the top right. Cancer diseases are Malignancies 

Figure 6; interesting shades, why 2 different shades in the controls , does that mean anything? if not you may want to keep it simple 

- Figures 2, 5 and 6 have been changed as requested by the Reviewers

Yet some additional cosmetic changes are recommended, for instance conclusions of abstract section the sentence is awkward even in spoken english . population  "whose 99 were males" .... Morphological data considering all eyes 103 had CME "whose  

- The language was reviewed by a native English speaker

Reviewer 3 Report

Comments and Suggestions for Authors

The study by Marcella Nebbioso et al. explored retinal morphology using optical coherence tomography (OCT) in a cohort of 307 patients affected with retinitis pigmentosa (RP). This is a single-center retrospective cross-sectional study, which focused primarily on three distinct OCT parameters, such as ellipsoid zone volume, thickness of the outer nuclear layer and subfoveal choroidal thickness. The authors described alterations in these parameters in studied population, as well as distribution of macular complications affecting RP patients, such as cystoid macular oedema, epiretinal membranes and vitreomacular traction. This is a 2nd review of this manuscript, in which I admit that some improvements were done in editing the manuscript and not in re-designing the study.

I suggest following improvements:

Broad comments  

1)      The title of this manuscript: I would change to “Macular alterations in the cohort of Caucasian patients affected with retinitis pigmentosa”

2)      The Abstract: conclusion is still very unclear. It should be the essence of main findings and value to the field need to be emphasized.

3)      The Introduction Section has been improved. I have few comments, as mentioned below in the “Specific comments” and few of them are related to linguistic problems (mainly grammar and structure), which also appear in the entire manuscript. I would suggest professional English language editor support before publishing this manuscript.  

4)      The Material and methods Section has been improved. Few comments suggesting improvement are presented under the “Specific comments”

5)      The Results section has been improved. However, the authors should be slightly more specific to present the data and emphasize main findings. I included my comments under the “Specific comments”.

6)      The Discussion section is better than before, however, still need to be edited and re-structured. For now, it is the weakest part of this manuscript. Few paragraphs are too long and findings are not clearly emphasized. I would recommend subtitles here describing main topics discussed (emphasized own findings in correlation to previously published data). What I have a concern here is that the results from this study are still very premature to formulate robust conclusion and avoid speculations. I refer to more specific comments below.

7)      Conclusions Section: need to be re-written. I refer to more specific comments below.

Specific comments

Abstract:

- row 18: macular disorders or vitreomacular disorders, or macular complications? Please, be specific in the entire manuscript.

- row 20: Start this statement with “To compare3 to be consistent with the first statement.

- row 33-35: this conclusion is not clear. The authors would need to specify their main findings and how they can contribute to the field

Introduction:

-row 51-63: this paragraph is irrelevant to this study, especially a part about syndromic forms of RP, which were excluded from the study

-row 95: change “searches” to “studies”

-row 97: change to “is also associated”

-row 99: macular disorders or vitreomacular disorders or macular complications? Change to appropriate one and be consistent in the entire manuscript

-row 102: “retinal dystrophies” or specifically RP here?

Material and methods:

-row 125: change to, e.g. A number of 307 patients with clinical and genetic diagnoses of RP were enrolled to this retrospective, observational and descriptive study

-row 150: change to healthy controls

-row 175: change “taken” to “obtained”

-row 179: place Enhanced Dept…. into brackets

Results:

-change the 1st statement to, e.g. “Re-write this statement: Total of 218 (436 eyes) patients of 307 patients initially enrolled were included for further morphological investigations of macula.”

-row 237: change subtitle to “OCT measurements”

-rows 249-253: improve grammar and structure of these sentences

-subtitle “Mixed complications” – change to, e.g. “A variety of macular alterations in RP patients”

-next subtitle, row 272: change to “Visual acuity in RP patients with different macular alterations”. Please, be specific while describing the results.

-row 288: improve this subtitle

Discussion:

This is the weakest part of the manuscript, and the reason for this is a study design and the results, which were obtained. The authors need to be very specific and careful while presenting own and limited data in order to emphasize them, and at the same time avoid speculations.

-row 318: macular disorders/ or macular alterations/ or vitreomacular disorders? Please, be specific.

-row 319: you need to specify which data you compared. Please avoid using “we compared some data”

-row 320-328: the content is relevant, however the structure and phrasing need to be improved

-row 329-359: the same here, the content is relevant, however the structure and phrasing need to be improved. Moreover, data reported by others are very long compared to description of own findings

- rows 360-383: this paragraph is also too long, a bit chaotic and does not correlate well with results of this manuscript

-rows 385-391: here, there is only one statement about own results. Emphasize your own results in relations to others

-rows 402-407: this section does not fit well here

-the 1st statement in the section “Study limitations and future directions” should be places in the context of conclusions. Please, start with an opening statement .

-row 414: vitreoretinal pathologies/ or macular disorders/ or macular alterations?

-rows 417-420: this statement is unnecessary

Conclusions

This section requires substantial editing in the context of findings from this manuscript, their scientific soundness in the context of already published data and future directions.

Figures and tables:

Figure 1 and 2: I think they may be not necessary to be presented since they do not directly refer to the results of this study. Figure 2 is presented in the Introduction Section, which is quite unusual.  Whereas, Figure 1 in the Methodology Section. Moreover, these two figures do not appear in the chronological order.

Current Figure 3: should be the first figure of this manuscript

Current Figure 5 and 6: are better now, however I would suggest a colour filling without any pattern for columns. The background of these figures should be white. It would increase the visibility.

Comments on the Quality of English Language

I would suggest the editing by the professional English language editor

Author Response

Dear Prof. Slawomir Teper

Guest Editor Diagnostics Journal,

and

Chair Clinical Department of Ophthalmology,

Faculty of Medical Sciences in Zabrze, Medical University of Silesia in Katowice, Poland

[Diagnostics] Special Issue: Optical Coherence Tomography Imaging in Retinopathy: New Advances and Future Trends - Submission Deadline: 30 September 2024 (e-mail 7/08/2024).

New title recommended by Reviewer 3: “Macular alterations in the cohort of caucasian patients affected with retinitis pigmentosa”

       We thank you for allowing us to send the revised manuscript titled “Macular alterations in the cohort of caucasian patients affected with retinitis pigmentosa” by Marcella Nebbioso, Elvia Mastrogiuseppe, Eleonora Gnolfo, Marco Artico, Antonietta Moramarco, Fabiana Mallone, Samanta Taurone, Annarita Vestri, and Alessandro Lambiase.

We are grateful for the useful comments and valuable improvements to our paper. Those changes are highlighted within the manuscript. Responses to the Reviewer comments have been reported at the bottom of the letter.

We hope that your journal will consider the manuscript.    

Best regards

Marcella Nebbioso and Co-Authors

Marcella Nebbioso, MD, Prof. (orcid.org/0000-0002-5512-0849)

e-mail: marcella.nebbioso@uniroma1.it Phone: ++39/06/49975422 Fax: ++39/06/49975426

Department of Sense Organs, Sapienza University of Rome, piazz.le A. Moro 5, 00185 Rome,

Italy.

REVIEWER 3

Comments and Suggestions for Authors

The study by Marcella Nebbioso et al. explored retinal morphology using optical coherence tomography (OCT) in a cohort of 307 patients affected with retinitis pigmentosa (RP). This is a single-center retrospective cross-sectional study, which focused primarily on three distinct OCT parameters, such as ellipsoid zone volume, thickness of the outer nuclear layer and subfoveal choroidal thickness. The authors described alterations in these parameters in studied population, as well as distribution of macular complications affecting RP patients, such as cystoid macular oedema, epiretinal membranes and vitreomacular traction. This is a 2nd review of this manuscript, in which I admit that some improvements were done in editing the manuscript and not in re-designing the study.

  • We thank the Reviewer for his help in improving the manuscript.

I suggest following improvements:

 Broad comments  

1)      The title of this manuscript: I would change to “Macular alterations in the cohort of Caucasian patients affected with retinitis pigmentosa”

- It was done

2)      The Abstract: conclusion is still very unclear. It should be the essence of main findings and value to the field need to be emphasized.

- It was done

3)      The Introduction Section has been improved. I have few comments, as mentioned below in the “Specific comments” and few of them are related to linguistic problems (mainly grammar and structure), which also appear in the entire manuscript. I would suggest professional English language editor support before publishing this manuscript.  

4)      The Material and methods Section has been improved. Few comments suggesting improvement are presented under the “Specific comments”

5)      The Results section has been improved. However, the authors should be slightly more specific to present the data and emphasize main findings. I included my comments under the “Specific comments”.

6)      The Discussion section is better than before, however, still need to be edited and re-structured. For now, it is the weakest part of this manuscript. Few paragraphs are too long and findings are not clearly emphasized. I would recommend subtitles here describing main topics discussed (emphasized own findings in correlation to previously published data). What I have a concern here is that the results from this study are still very premature to formulate robust conclusion and avoid speculations. I refer to more specific comments below.

7)      Conclusions Section: need to be re-written. I refer to more specific comments below.

Specific comments

Abstract:

- row 18: macular disorders or vitreomacular disorders, or macular complications? Please, be specific in the entire manuscript.

- The words have been standardized in the paper “macular complications”

- row 20: Start this statement with “To compare3 to be consistent with the first statement.

- It was done

- row 33-35: this conclusion is not clear. The authors would need to specify their main findings and how they can contribute to the field

- It was done

Introduction:

-row 51-63: this paragraph is irrelevant to this study, especially a part about syndromic forms of RP, which were excluded from the study

- The paragraph has been deleted

-row 95: change “searches” to “studies”

-row 97: change to “is also associated”

- It was done

-row 99: macular disorders or vitreomacular disorders or macular complications? Change to appropriate one and be consistent in the entire manuscript

- The words have been standardized in the paper “Macular complications”

-row 102: “retinal dystrophies” or specifically RP here?

- The word has changed to RP

Material and methods:

-row 125: change to, e.g. A number of 307 patients with clinical and genetic diagnoses of RP were enrolled to this retrospective, observational and descriptive study

- The sentence has changed as requested by the Reviewer

-row 150: change to healthy controls

-row 175: change “taken” to “obtained”

-row 179: place Enhanced Dept…. into brackets

- The corrections have been made

Results:

-change the 1st statement to, e.g. “Re-write this statement: Total of 218 (436 eyes) patients of 307 patients initially enrolled were included for further morphological investigations of macula.”

- The sentence has changed as requested by the Reviewer

-row 237: change subtitle to “OCT measurements”

- The words have been changed

-rows 249-253: improve grammar and structure of these sentences

- The sentences have been changed

-subtitle “Mixed complications” – change to, e.g. “A variety of macular alterations in RP patients”

-next subtitle, row 272: change to “Visual acuity in RP patients with different macular alterations”. Please, be specific while describing the results.

-row 288: improve this subtitle

 -The subtitles has changed

Discussion:

This is the weakest part of the manuscript, and the reason for this is a study design and the results, which were obtained. The authors need to be very specific and careful while presenting own and limited data in order to emphasize them, and at the same time avoid speculations.

-row 318: macular disorders/ or macular alterations/ or vitreomacular disorders? Please, be specific.

- The words have been changed macular complications

-row 319: you need to specify which data you compared. Please avoid using “we compared some data”

-The sentence has been changed

-row 320-328: the content is relevant, however the structure and phrasing need to be improved

-row 329-359: the same here, the content is relevant, however the structure and phrasing need to be improved. Moreover, data reported by others are very long compared to description of own findings

- rows 360-383: this paragraph is also too long, a bit chaotic and does not correlate well with results of this manuscript

-rows 385-391: here, there is only one statement about own results. Emphasize your own results in relations to others

-All changes requested by the Reviewer have been made to the manuscript.

-rows 402-407: this section does not fit well here

- The section has been moved

-the 1st statement in the section “Study limitations and future directions” should be places in the context of conclusions. Please, start with an opening statement.

- The suggested changes have been made

-All changes requested by the Reviewer have been made to the manuscript.

-row 414: vitreoretinal pathologies/ or macular disorders/ or macular alterations?

- Changed to macular alterations

-rows 417-420: this statement is unnecessary

- The statement on lines 417-420 and 2 related bibliographic references were requested by the Reviewer 2 of the first revision, and we have complied with both of the Reviewer's requests.  Reviewer 2

(1) https://doi.org/10.1016/j.omtn.2022.04.015

(2) https://doi.org/10.3390/diagnostics14070764

Conclusions

This section requires substantial editing in the context of findings from this manuscript, their scientific soundness in the context of already published data and future directions.

- The section has been revised

Figures and tables:

Figure 1 and 2: I think they may be not necessary to be presented since they do not directly refer to the results of this study. Figure 2 is presented in the Introduction Section, which is quite unusual.  Whereas, Figure 1 in the Methodology Section. Moreover, these two figures do not appear in the chronological order.

Current Figure 3: should be the first figure of this manuscript

- The position of the figures has been changed; the figures are all our patients and healthy subjects.

- The figures and measurements reported relate to our research. The normal figure was requested by a Reviewer 1 in the first revision, so we have included it as requested.

Current Figure 5 and 6: are better now, however I would suggest a colour filling without any pattern for columns. The background of these figures should be white. It would increase the visibility.

- Figures 5 and 6 have been changed as requested by the Reviewer.

Round 3

Reviewer 3 Report

Comments and Suggestions for Authors

The study by Marcella Nebbioso et al. explored retinal morphology using optical coherence tomography (OCT) in a cohort of Caucasian patients affected with retinitis pigmentosa (RP). This is a single-center retrospective cross-sectional study, which focused primarily on three distinct OCT parameters, such as ellipsoid zone volume, thickness of the outer nuclear layer and subfoveal choroidal thickness. The authors described alterations in these parameters in studied population, as well as distribution of macular complications affecting RP patients, such as cystoid macular oedema, epiretinal membranes and vitreomacular traction. This is the third review of this manuscript.

Broad comments  

1)      The comments were addressed in editing the manuscript and not in re-designing the study.

2)      In general, the manuscript has been significantly improved, however the Discussion Section and Conclusions still require improvement. Both sections appears unstructured, slightly chaotic and are difficult to follow.

3)      I would also recommend a final editing by linguistic English language professional since phrasing need to be improved.

Specific comments

Abstract:

- row 33: please change to “in a cohort of patients affected with RP, the most common form of inherited retinal degenerations worldwide”-row 34: change to: “The results of our study indicate the importance of regular monitoring of RP patients and early intervention to avoid further complications in this already visually impaired group of patients”

Introduction:

-row 46: remove a word “noticed”

-row 48: write “in European population”

-row 52-53: perhaps remove this statement since it is irrelevant to this study

-row 75: remove “to occur”

-row 82: change “persist” to “is”

-row 83: change this statement and start with “In the contrary, many studies have shown retinal thinning in RP patients….”

-row 86: change “In addition” to “Furthermore, it was shown that…”

Discussion:

-row 307: change the beginning of this statement to as follows: Previous studies showed that…” and add References here.

-row 312: change “and colleagues” to “et al.”

-row 316-317: write clear relation “ONLT was reduced/or increased in the group of…”

-row 321: begin the statement with “The studies shown by Guo at al. showed…”

-row 335: change to “We propose….”, then change “suffering from3 to 3having well-defined macular alterations….”

-row 341: change the phrase “In essence” to another opening paragraph phrase

-row 344: change phrasing here “Pachychoroid may be regarded as a clinical entity….”

-row 351-354: improve the entire statement

-row 355-357: please, remove this statement

-Subsection Study limitations and future directions requires significant improvement in the context of concept and structure.

Row 377-378: remove this statement; this is the aim of your study.

Row 380-386: please, remove this paragraph or substantially edit it.

Row 387-389, 400-404: this is the fact and not limitation or future direction

Row 390-392: Write about limitations of single-center study vs. multi-center studies

Row 404-406, 407-409: please, remove or edit in the context of what is your take on this. Be specific and focused, so that future directions of this study are connected to the main findings and what can be done in the near future and not overall in the field of IRDs.

Conclusions:

-row 412: change phrasing in the 1st statement of this Section

-row 415: edit please since Luxturna is available for RPE65-RP and agnostic therapies as well. Be very careful with optimal phrasing in this statement

-row 418-422: remove or write in your own words. The conclusions are your owns and do not include ref.

Comments on the Quality of English Language

  I would also recommend a final editing by linguistic English language professional since phrasing need to be improved. I strongly believe it would benefit the authors and their work. 

Author Response

20 October, 2024

Dear Prof. Slawomir Teper

Guest Editor Diagnostics Journal,

and

Chair Clinical Department of Ophthalmology,

Faculty of Medical Sciences in Zabrze, Medical University of Silesia in Katowice, Poland

[Diagnostics] Special Issue: Optical Coherence Tomography Imaging in Retinopathy: New Advances and Future Trends - Submission Deadline: 30 September 2024 (e-mail 7/08/2024).

New title recommended by Reviewer 3: “Macular alterations in the cohort of caucasian patients affected with retinitis pigmentosa”

       We thank you for allowing us to send the revised manuscript titled “Macular alterations in the cohort of caucasian patients affected with retinitis pigmentosa” by Marcella Nebbioso, Elvia Mastrogiuseppe, Eleonora Gnolfo, Marco Artico, Antonietta Moramarco, Fabiana Mallone, Samanta Taurone, Annarita Vestri, and Alessandro Lambiase.

We are grateful for the useful comments and valuable improvements to our paper. Those changes are highlighted within the manuscript. Responses to the Reviewer comments have been reported at the bottom of the letter.

We hope that your journal will consider the manuscript.    

Best regards

Marcella Nebbioso and Co-Authors

Marcella Nebbioso, MD, Prof. (orcid.org/0000-0002-5512-0849)

e-mail: marcella.nebbioso@uniroma1.it Phone: ++39/06/49975422 Fax: ++39/06/49975426

Department of Sense Organs, Sapienza University of Rome, piazz.le A. Moro 5, 00185 Rome,

Italy.

REVIEWER 3

Comments and Suggestions for Authors

The study by Marcella Nebbioso et al. explored retinal morphology using optical coherence tomography (OCT) in a cohort of Caucasian patients affected with retinitis pigmentosa (RP). This is a single-center retrospective cross-sectional study, which focused primarily on three distinct OCT parameters, such as ellipsoid zone volume, thickness of the outer nuclear layer and subfoveal choroidal thickness. The authors described alterations in these parameters in studied population, as well as distribution of macular complications affecting RP patients, such as cystoid macular oedema, epiretinal membranes and vitreomacular traction. This is the third review of this manuscript.

 Broad comments  

1)      The comments were addressed in editing the manuscript and not in re-designing the study.

2)      In general, the manuscript has been significantly improved, however the Discussion Section and Conclusions still require improvement. Both sections appears unstructured, slightly chaotic and are difficult to follow.

3)      I would also recommend a final editing by linguistic English language professional since phrasing need to be improved.

- We thank the Reviewer for his help in improving the manuscript.

Specific comments

Abstract:

- row 33: please change to “in a cohort of patients affected with RP, the most common form of inherited retinal degenerations worldwide”-row 34: change to: “The results of our study indicate the importance of regular monitoring of RP patients and early intervention to avoid further complications in this already visually impaired group of patients”

 - The words have been changed as requested by the Reviewer

Introduction:

-row 46: remove a word “noticed”

-row 48: write “in European population”

-row 52-53: perhaps remove this statement since it is irrelevant to this study

-row 75: remove “to occur”

-row 82: change “persist” to “is”

-row 83: change this statement and start with “In the contrary, many studies have shown retinal thinning in RP patients….”

-row 86: change “In addition” to “Furthermore, it was shown that…”

 - The words and sentences have been changed as requested by the Reviewer

Discussion:

-row 307: change the beginning of this statement to as follows: Previous studies showed that…” and add References here.

-row 312: change “and colleagues” to “et al.”

-row 316-317: write clear relation “ONLT was reduced/or increased in the group of…”

-row 321: begin the statement with “The studies shown by Guo at al. showed…”

-row 335: change to “We propose….”, then change “suffering from3 to 3having well-defined macular alterations….”

-row 341: change the phrase “In essence” to another opening paragraph phrase

-row 344: change phrasing here “Pachychoroid may be regarded as a clinical entity….”

-row 351-354: improve the entire statement

-row 355-357: please, remove this statement

  - The words and sentences have been changed as requested by the Reviewer

-Subsection Study limitations and future directions requires significant improvement in the context of concept and structure.

Row 377-378: remove this statement; this is the aim of your study.

Row 380-386: please, remove this paragraph or substantially edit it.

Row 387-389, 400-404: this is the fact and not limitation or future direction

Row 390-392: Write about limitations of single-center study vs. multi-center studies

Row 404-406, 407-409: please, remove or edit in the context of what is your take on this. Be specific and focused, so that future directions of this study are connected to the main findings and what can be done in the near future and not overall in the field of IRDs.

  - The words and sentences have been changed as requested by the Reviewer

Conclusions:

-row 412: change phrasing in the 1st statement of this Section

-row 415: edit please since Luxturna is available for RPE65-RP and agnostic therapies as well. Be very careful with optimal phrasing in this statement

-row 418-422: remove or write in your own words. The conclusions are your owns and do not include ref.

 - The words and sentences have been changed as requested by the Reviewer